# The Association Between the Changes in General, Family, and Financial Aspects of Quality of Life and Their Effects on Cognitive Function in an Elderly Population: The Korean Longitudinal Study of Aging, 2008–2016

**DOI:** 10.3390/ijerph17031106

**Published:** 2020-02-10

**Authors:** Wonjeong Chae, Eun-Cheol Park, Sung-In Jang

**Affiliations:** 1 Department of Public Health, College of Medicine, Yonsei University, Seoul 03722, Korea; wjchae0816@yuhs.ac; 2Institute of Health Services Research, Yonsei University, Seoul 03722, Korea; ecpark@yuhs.ac; 3Department of Preventive Medicine, Yonsei University College of Medicine, Seoul 03722, Korea

**Keywords:** aging, quality of life, cognitive decline, Mini-Mental State Examination, gender differences, Korea

## Abstract

*Background* The growing aging population is a global phenomenon and a major public health challenge. Among Organization for Economic Co-operation and Development countries, Korea is the fastest aging country. We aimed to investigate the relationship between changes in quality of life (QOL) and cognitive function in older adults. Method: Data from the Korean Longitudinal Study of Aging collected from 2008 to 2016 were used. In 3453 participants (men: 1943; women: 1541), QOL was measured by three aspects: general, financial, and familial. Changes in QOL status were assessed by four categories: remained poor, worsened, improved, and remained good. The level of cognitive function was measured by the Mini-Mental State Examination score (MMSE, normal range cut-off value: 24 or above). For the statistical analysis, the generalized equation model (GEE) was performed. Results: For all three aspects of QOL measured, participants whose QOL score remained poor were associated with cognitive decline that their odds ratios (OR) were statistically significant (general: OR = 1.33; familial: OR = 1.39; financial: OR = 1.40). For subgroup analysis by gender, the highest OR in men was the financial aspect of QOL (OR = 1.45); in women, the highest OR was the familial aspect of QOL (OR = 1.75). Conclusion: This study showed an association between QOL and cognitive function in a Korean elderly population. Our findings suggest that QOL measurements with a gender-specific approach can be used as a tool to detect cognitive changes in older adults and help prevent or delay cognitive decline.

## 1. Introduction

The growing aging population has become a global issue that reflects changes in society as modernization progresses in nearly all countries [1,2]. To help tackle this issue, the World Health Organization (WHO) has emphasized that countries promote healthy aging. According to a report by the United Nations, 382 million people worldwide were aged 60 years or older in 1980, a number which has grown to more than 962 million people in 2017, and is predicted to be close to 2.1 billion by 2050 [3]. As defined by WHO, an aging society is a society in which the proportion of people aged 65 years and older accounts for more than 7% of the total population; an aged society, more than 14% of the population; and super-aged society, more than 20% of the population [4,5]. In recent decades, the life expectancy in most countries has increased [3,5,6,7]. In particular, advanced technology and improved access to healthcare has greatly contributed to lowering the mortality rate, extending life expectancies, and improving quality of life. However, many societies with growing aging populations have been simultaneously experiencing another major health issue: low birth rate [1,3,6,8].

South Korea is the fastest aging society among Organization for Economic Co-operation and Development (OECD) countries [9]. South Korea was recently reclassified as an aged society, because the elderly population increased by 14.3% in 2018 [9]. South Korea became an aging society in 18 years and an aged society in 8 years, while the United States took 73 years and 21 years, Germany took 40 years and 37 years, and Japan took 24 years and 12 years [1,10,11].

An increase in the elderly population of a society leads to complex problems related to social, economic and health issues [6,7,10,12]. Concomitantly, a decrease in the young population reduces work participation and industrial productivity, resulting in economic decline [6,8]. Furthermore, these changes increase the financial burden of countries. In addition, that a country’s welfare system struggles to care for older adults as the aging rate grows [3,6,8]. Although lifespan has increased, a growing aging population is now faced with problems such as poverty, disease, and reclusion [4,6,13]. Even when the health of older adults improves, their use of medical care increases, leading to a rise in poverty among this population [3,5]. Changes in family composition and rapid social change are believed to be the reasons for the rising occurrence of countries with rapidly growing elderly populations [6,8,14]. In particular, countries with rapid population changes are expected to encounter these problems on a massive scale, requiring the emergence of aging measurements and welfare policies for older adults [1,3].

During the aging progress, changes in the physical and mental health status of a patient may not be considered a major issue until they are diagnosed with a disease or disorder, such as arthritis, dementia, and Alzheimer’s disease, because these changes are generally considered to be a part of aging [15]. Cognitive function determines an individual’s ability to complete information transformation, decision making, and daily activities [15,16]. Previous studies have associated cognitive function with chronic diseases such as diabetes, hypertension, and obstructive pulmonary disease [17,18,19,20]. However, cognitive function is also related to socioeconomic factors such as resident area, occupation, and education level that can influence quality of health [21,22].

The WHO defined health by three elements: physical, mental and social well-being [23]. Those elements are also closely related to quality of life [24]. So, quality of life could be one of measurement for health. In older adults, quality of life can have an overall association on the deterioration of cognitive function, including personal, familial, and social factors [24,25]. In line with that, quality of life can be expressed as the mixture of intrapersonal and social normative elements in past, current, and future [13,20,25,26]. Among the elderly population, quality of life showed a linkage with their cognitive level [27,28]. A study conducted by Woods et al. reported that among dementia patients, improving patients’ quality of life affected positively in their treatment and improving their cognitive function [27].

Therefore, the deterioration of an individual’s quality of life may negatively affect their cognitive function. Many existing papers describe the relationship between the life expectancy of patients and their quality of life due to cognitive decline [2,15,17,18]. However, these studies have not suggested methods to use in preparation of an aging or aged society by monitoring quality of life. In addition, previous studies were investigated on quality of life as a health-related factor [29,30,31]. The purpose of this study was to investigate the effect of the quality of life on the cognitive function in a Korean elderly population. By measuring overall quality of life, as well as its familial-specific and financial-specific aspects, this study aimed to provide a tool for the prevention and early identification of cognitive decline among older adults.

## 2. Methods

### 2.1. Data Source and Study Population

This study used data of the Korean Longitudinal Study of Aging (KLoSA) from 2008 to 2016. KLoSA is a representative data source that was conducted by the Korea Employment Information Service (KEIS) on behalf of the South Korean Ministry of Employment and Labor. The data provides various aspects of Korean elderly population and is comparable to the Health and Retirement Study (HRS) in the USA—as well as the Survey of Health, Aging, and Retirement in Europe (SHARE); the Japanese Study of Aging and Retirement (JSTAR); and the Chinese Health and Retirement Longitudinal Study (CHARLS). The survey is based on the multistage stratified design among Korean adults aged 45 years or older in 2006. Approximately 10,000 Korean adults were interviewed biannually using computer-assisted personal interviewing methods, and the latest data available is for 2016 [32].

Participants who had low cognitive levels or dementia based on an MMSE score less than 23 were eliminated in 2006. the baseline year was 2008 when a total of 4463 participants were over 45 years of age. To measure the familial aspect of quality of life, our study population was limited to 3453 participants who were married and had at least one child. There were 1943 men and 1541 women included.

### 2.2. Quality of Life

For this study, quality of life was measured in three categories: general aspects, financial aspects and familial aspects. Similar to the EuroQOL visual analogue scale (as EQ-VAS), participants provided the exact score for each category from 0 to 100. If a score was below the mean score, then it was defined as ‘bad’; if a score was equal or above the mean score, then it was defined as ‘good’. We applied a lag-time option to detect the changes in quality of life compared to the prior year. Therefore, we grouped the scores into four categories: (1) remained poor, (2) worsened, (3) improved, and (4) remained good.

### 2.3. Cognitive Level, Mini-Mentel State Examination (MMSE)

Cognitive function was measured by the Mini-Mental State Examination (MMSE) score. The MMSE was developed by Folstein et al. to evaluate cognitive function according to seven items [33] that measure time awareness, place awareness, a record of three words, care and calculation, a memory of three words, language and constructive visual capacity [33,34]. The maximum score for this test is 30, signifying peak cognitive function. Generally, a score above 24 is considered normal. A score less than 18 denotes moderate cognitive impairment, suggesting the individual is at risk of dementia; a score between 18 to 23 denotes mild cognitive impairment, indicating cognitive decline [33]. For the study population, we excluded participants who scored less than 23 in 2006 to control for low cognitive level. As the outcome variable, we grouped cognitive function into two groups, low and normal.

### 2.4. Covariates

Factors were adjusted to control baseline variables that can have a possible effect on our study outcome and to improve the precision of the outcome. Therefore, in our study, factors related to demographics (sex, age), socioeconomics (education level, income level, employment status, marriage status, social activity: frequently—almost every day, once a week, twice a week, thrice a week; often—once a month, twice a month, once a two month; rarely—once a year, twice year, thrice a year), family (number of children, degree of face-to-face contact with children, degree of email or phone contact with children: frequently—almost every day, once a week, twice a week, thrice a week; often—once a month, twice a month, once a two month; rarely—once a year, twice year, thrice a year), health behavior (smoking, drinking) and health condition that diagnosed by their doctor (number of chronic diseases, cancer, depression) were adjusted for the study. All categories are grouped based on the survey question for those without further definition.

### 2.5. Statistical Analyses

Descriptive analyses were conducted, and chi-square tests were performed to investigate an association between the changes in quality of life from the prior year and cognitive function. To analyze the longitudinal and repeated measures data using a logistic regression would violate the assumption of independence. Therefore, we conducted a multivariable generalized estimating equation (GEE) model with a logit link and controlled for confounders. Presented outcomes show odds ratio and 95% confidence intervals. The software SAS 9.4 (SAS Institute, Cary, NC, USA) was used for this study.

## 3. Results

The characteristics of the study population in baseline year 2008 are shown in Table 1. Of the 3453 participants, 14.7% (*n* = 522) were classified with low cognitive function at baseline, and 82.7% (*n* = 2931) were classified with normal cognitive function. For the general aspect of quality of life, 20.2% of participants (*n* = 698) remained poor, 11.7% (*n* = 405) worsened, 22.2% (*n* = 768) improved, and 45.8% (*n* = 1582) remained good. For the familial aspect of quality of life, 25.7% of participants (*n* = 888) remained poor, 28.4% (*n* = 981) worsened, 11.7% (*n* = 403) improved, and 34.2% (*n* = 1181) remained good. For the financial aspect of quality of life, 24.1% of participants (*n* = 832) remained poor, 6.2% (*n* = 218) worsened, 32.2% (*n* = 1111) improved, and 37.4% (*n* = 1292) remained good at baseline, 31.9% of participants had assessed their general quality of life as bad; 54.5%, familial quality of life; and 30.4%, financial quality of life. By contrast, 68.1% of participants had assessed their general quality of life as good; 45.9%, familial quality of life; and 69.6%, financial quality of life.

The results of using a GEE model with the logit link to assess the association between three aspects of quality of life and cognitive level are expressed in Table 2. With the ‘remained good’ group used as a reference, participants who either ‘remained poor’, ‘worsened’, or ‘improved’ show higher ORs of low cognitive function in all three categories of quality of life after adjusting for covariates. In addition, participants who ‘improved’ had higher ORs (odds ratios) than the reference group. For general quality of life, the OR was 1.33 (95% CI: 1.18–1.51) for the ‘remained poor’ group, 1.32 (95% CI: 1,18–1.47) for the ‘worsened’ group, and 1.16 (95% CI: 1.01–1.30) for the ‘improved’ group. For familial quality of life, the OR was 1.39 (95% CI: 1.23–1.58) for the ‘remained poor’ group, 1.33 (1.18–1.50) for the ‘worsened’ group, and 1.19 (95% CI: 1.04–1.35) for the ‘improved’ group. For financial quality of life, the OR was 1.40 (95% CI: 1.23–1.58) for the ‘remained poor’ group, 1.37 (95% CI: 1.21–1.55) for the ‘worsened’ group, and 1.25 (95% CI: 1.11 - 1.41) for the ‘improved’ group.

Subgroup analysis presents associations between quality of life and low cognitive level within the gender and the employment status of study the population. Figure 1 and Figure 2 presents the results of the subgroup analysis. Three aspects of quality of life are grouped individually and contain the degree of quality of life: remained poor, worsen, and improve, respectively.

Figure 1 contains the results of the subgroup analysis conducted by gender that shows low cognitive function measured by the MMSE score. Men and women express on different aspects of quality of life regarding their cognitive function. For men, the financial quality of life was the most affected aspect of quality of life, followed by the general aspect and the family aspect. Financial aspect had the OR of 1.44 (95% CI: 1.19–1.74), 1.45 (95% CI: 1.20–1.71), and 1.35 (95% CI: 1.13–1.60) as the categorical orders: remained poor, worsened, and improved. The ‘worsened’ group had the highest OR of 1.36 (95% CI: 1.13–1.62) of all groups for general quality of life. In addition, the ‘worsened’ group had the highest OR of 1.18 (95% CI: 1.00–1.39) of all groups for familial quality of life.

Figure 2 shows the results of subgroup analysis that was conducted on employment status. The results demonstrated that participants who were unemployed had higher ORs of low cognitive function measured by the MMSE score. For general quality of life, the unemployed group had an OR of 1.37 (95% CI: 1.19–1.59), 1.37 (95% CI: 1.19–1.56), and 1.15 (95% CI: 1.00–1.33) for the categorical orders remained poor, worsened, and improved, respectively. For the financial aspect, the unemployed group had an OR of 1.49 (95% CI: 1.28–1.73), 1.49 (95% CI: 1.28–1.73), and 1.31 (1.13–1.51). For the family aspect, the unemployed group had an OR of 1.42 (95% CI: 1.23–1.65), 1.35 (95% CI: 1.17–1.57), and 1.31 (1.05–1.45). The expressed results are reported in categorical order with remained good as the reference group.

## 4. Discussion

This study was designed to investigate the effect of different aspects of quality of life on the cognitive function of the elderly population in South Korea. Our findings confirmed that poor quality of life in the general, familial, and financial aspects showed associations with the risk of cognitive decline. Further subgroup analysis identified a gender difference: men had a strong association of quality of life with financial aspects, while women had a strong association between quality of life and familial aspects.

An individual’s quality of life largely affects their daily living, physical health, and mental health [13]. When this measure remains good for several years, then the individual has a higher likelihood of achieving good health as an older adult [13,26]. Furthermore, having a good quality of life can lead to healthy aging. Similar to Saracli et al. and Helvik et al., we identified an association between quality of life and cognitive function [20,25]. Similar to the previous studies [5,20,25], our study found that participants who have a good quality of life for years have the least risk of low cognitive function, whereas those who have a poor quality of life for years, or a worsening quality of life, had the highest or a higher risk of low cognitive function. An interesting factor was found in the group of people whose quality of life had improved. When compared to the prior year, an improved quality of life score still showed a greater risk of low cognitive function. Lamourex-Lamarche and Vasiliadis discovered that once a person experiences an event that is significant and traumatic, then it may remain so long after the event concludes and have a long-term cognitive effect [35]. This event also affects quality of life [35] in our study, who experienced poor quality of life once could be considered as had a traumatic event.

Our analyses support the theory of gender as a different mechanism in mental health [36,37] which can be presented as the gender effect. The study population includes individuals who are aged 50 to 70 and above. Therefore, there might be cohort differences that shared different social and non-social events throughout the aging process. Despite the cohort effect on the elderly population, gender differences are easily found in a society that can possibly affect the quality of life and cognitive level [38,39,40,41]. Men and women tend to participate in different social activities which could influence their quality of life and/or cognitive level [39,40]. With regard to cognitive level, gender has a stronger association. Many studies have shown that women are at a higher risk of dementia and Alzheimer’s disease [14,40,41], and their study subjects were the elderly population of various ages. Their results were consistent with the investigations conducted on gender difference on dementia and Alzheimer’s disease among elders [14,40,41,42,43], and our results exhibited a higher OR of the low cognitive function groups in women. In addition, our study evaluated various aspects of quality of life and found that the effects of these aspects are different by gender [24,31,38] which is also interpreted as one part of gender effect. Men, who are traditionally in charge of family finances, were more sensitive to the financial aspect of quality of life. By contrast, women were more vulnerable to the family aspect of quality of life, corresponding to their traditional roles centered around family [38]. Similar to our study, Campos et al. reported that quality of life was associated with physical health for men and psychosocial health for women, and especially for men, this measure showed a higher association with socioeconomic status [44].

Other studies have shown that cognitive function is related to educational levels [16,45]. Similar to these studies, which found that educated people are less likely to experience cognitive decline in early age, our study demonstrated that the higher the educational level, the lower the riskca of cognitive decline. In addition, older adults who hold a part-time job or volunteering work after retirement have been found to have better mental health [15,16]. Similarly, our study found that participants with a job were less likely to have low cognitive function.

Emerging aging societies are a global phenomenon where health issues related to cognitive decline are expected [3,6,15]. Generally, cognitive decline is a natural process of aging that all aging adults expect to face. Cognitive decline is noticed when changes occur in an individual’s information transformation, processing speed, and problem-solving activities [3,41,42,43,44]. Cognitive decline is a risk aspect of mild cognitive impairment (MCI) that can cause mental illnesses, such as dementia or Alzheimer’s disease. Although the biological mechanism of cognitive decline is unavoidable [41,45], previous studies have discovered that such non-pharmacological lifestyle strategies such as physical activities, healthy diet, music therapies, and cognitive training can delay cognitive decline [16,46,47,48,49,50,51]. These studies stress the importance of quality of life, especially in an elderly population. Older adults are susceptible to declining quality of life as they go through several major life events, including losing physical and mental capabilities, exiting from job market, and experiencing isolation [1,5,6,8,13]. Aspects related to declining quality of life in older adults can expedite the manifestation of cognitive problems [13,15,16,25].

Although individuals must adjust to the transformation of aging, their society must prepare for these changes as well. To obtain a healthy aging/aged society, a social institution to support these individuals must be established [1,3,5,6,8]. The previous study [2,5,6,11] expressed all recommendations that societies should implement. First, they indicated that a decline in the workforce leads to a loss of national competitiveness. Second, social welfare issues such as pension, medical costs, residential care, and other benefits should be discussed. Third, medical costs to provide care to older adults are a huge burden on countries. Among various medical services, costs related to mental health are high and growing every year [52]. according to South Korea’s Ministry of Health and Welfare, 105,598 people were diagnosed with MCI in 2014, which increased from 24,602 in 2010—an almost 4.3-fold increase within 5 years [53]. In the UK, 40% of hospital admissions are due to cognitive failure [15]. Besides South Korea and the UK, several countries are facing in increase in the number of mental health patients, especially those in cognitive decline [2,6,51,52,53,54,55]. In addition, the care for cognitive decline is expensive and causes massive financial burden on the individual and society. In the case of South Korea, the Ministry of Health and Welfare announced that the government paid KRW 35 billion (USD 31 million) to MCI patients in 2014, which was 52% higher than the medical costs of 2010 for patients aged 40 years or older [53]. Thus, managing patients with MCI would be an essential aspect to safe medical expenditure.

There are several strengths to our study. First, we analyzed the quality of life in different aspects and discovered gender differences among these aspects. A method of EQ-VAS was not used for measuring quality of life; however, the process of measurement used the same 0–100 score range. Second, the data source was from a Korean study administering a survey that obtained sufficient validity to a large sample size during a 10-year study period. Furthermore, it is comparable to similar data internationally, such as the HRS, SHARE, JSTAR and CHARLS. Third, the study emphasized the assets of longitudinal data by capturing changes in quality of life.

Despite of several strengths, this study has limitations. First, our population included only the participants who were married with at least one child to measure the familial aspect of quality of life. Even though we included participants who were widowed or divorced, this decision excluded a certain portion of the population. Second, we did not consider the health aspect of quality of life and measure its association with cognitive function because cognitive function could be influenced by the quality of health status. Third, to measure changes in quality of life, we used the mean of each aspect for the cut-off score of the self-reported data. Therefore, the measurement could be considered not as scientific a method; however, it was acceptable to detect the changes in quality of life. Fourth, the MMSE score is not the most accurate and efficient indicator to measure the cognitive level [56]. However, it has been commonly used and considered as a benchmark for newer measurements [57]. As our data is from the survey, we chose the variable with external validity instead of self-reported. Finally, our study design cannot show the causal relationship between changes in quality of life and cognitive level and we might not include all possible confounders for the study to adjust.

## 5. Conclusions

We investigated the association between quality of life and cognitive function. In addition, we identified gender-specific associations of quality of life with cognitive function. As many countries are reaching the status of an aging society, providing adequate social welfare to aging populations—including healthcare—is important. Therefore, our study suggests that quality of life metrics should be used as a tool to detect cognitive changes in older adults, to prevent or delay cognitive decline. Policymakers should consider our results to develop supportive programs for older adults to promote healthy aging.

## Figures and Tables

**Figure 1 ijerph-17-01106-f001:**
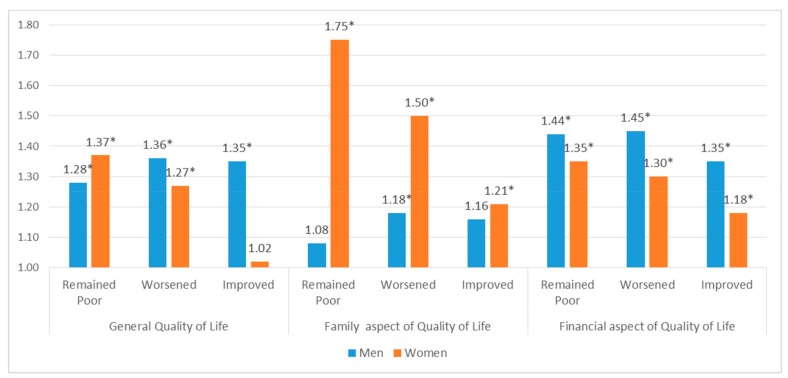
Subgroup analysis on gender: qualitive life with low cognitive function † measured by MMSE ‡ score. † Low: MMSE ≤ 23, normal: 24 ≤ MMSE; ‡ Mini- Mental State Examination (MMSE); * Statistically significant value.

**Figure 2 ijerph-17-01106-f002:**
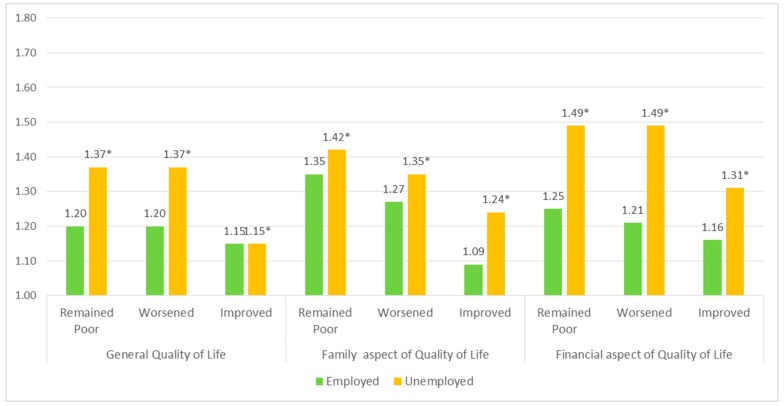
Subgroup analysis on employment status: qualitive life with low cognitive function † measured by MMSE ‡ score. † Low: MMSE ≤ 23, normal: 24 ≤ MMSE; ‡ Mini- Mental State Examination (MMSE); * Statistically significant value.

**Table 1 ijerph-17-01106-t001:** General characteristics of the study population in 2008.

Variables		Total (*N* = 3453)		Cognitive Function † Measured by MMSE Score ‡				*p*-Value *
				Low (*N* = 522)		Normal (*N* = 2931)		
		*N*	%	*N*	%	*N*	%	
General Quality of Life (QOL)								<0.0001
	Remained poor	698	20.2	144	20.6	554	79.4	
	Worsened	405	11.7	74	18.3	331	81.7	
	Improved	768	22.2	119	15.5	649	84.5	
	Remained good	1582	45.8	185	11.7	1397	88.3	
Family aspect of QOL								<0.0001
	Remained poor	888	25.7	207	23.3	681	76.7	
	Worsened	981	28.4	115	11.7	866	88.3	
	Improved	403	11.7	58	14.4	345	85.6	
	Remained good	1181	34.2	142	12.0	1039	88.0	
Financial aspect of QOL								<0.0001
	Remained poor	832	24.1	182	21.9	650	78.1	
	Worsened	218	6.3	54	24.8	164	75.2	
	Improved	1111	32.2	156	14.0	955	86.0	
	Remained good	1292	37.4	130	10.1	1162	89.9	
General QOL (baseline)								<0.0001
	Bad	1103	31.9	218	19.8	885	80.2	
	Good	2350	68.1	304	12.9	2046	87.1	
Financial aspects of QOL (baseline)								<0.0001
	Bad	1050	30.4	236	22.5	814	77.5	
	Good	2403	69.6	286	11.9	2117	88.1	
Family aspects of QOL (baseline)								0.0002
	Bad	1869	54.1	322	17.2	1547	82.8	
	Good	1584	45.9	200	12.6	1384	87.4	
**Sex**								<0.0001
	Men	1943	56.3	216	11.1	1727	88.9	
	Women	1510	43.7	306	20.3	1204	79.7	
Age								<0.0001
	50–59	1126	32.6	63	5.6	1063	94.4	
	60–69	1338	38.7	178	13.3	1160	86.7	
	70 or above	989	28.6	281	28.4	708	71.6	
Education								<0.0001
	Primary	1462	42.3	367	25.1	1095	74.9	
	Secondary	675	19.5	72	10.7	603	89.3	
	Tertiary	955	27.7	73	7.6	882	92.4	
	Beyond Tertiary	361	10.5	10	2.8	351	97.2	
Income level								<0.0001
	1Q	663	19.2	150	22.6	513	77.4	
	2Q	798	23.1	189	23.7	609	76.3	
	3Q	990	28.7	133	13.4	857	86.6	
	4Q	1002	29.0	50	5.0	952	95.0	
Employment Status								<0.0001
	Employed	1610	46.6	145	9.0	1465	91.0	
	Unemployed	1843	53.4	377	20.5	1466	79.5	
Marriage								<0.0001
	Currently Married	2867	83.0	372	13.0	2495	87.0	
	Once Married	586	17.0	150	25.6	436	74.4	
Number of children								<0.0001
	1	167	4.8	15	9.0	152	91.0	
	2	1118	32.4	95	8.5	1023	91.5	
	3	1078	31.2	143	13.3	935	86.7	
	4 or more	1090	31.6	269	24.7	821	75.3	
Meeting with children								0.4961
	Frequently	652	18.9	89	13.7	563	86.3	
	Often	254	7.4	38	15.0	216	85.0	
	Rarely/Never	2547	73.8	395	15.5	2152	84.5	
Phone call or email with children								0.1033
	Frequently	869	25.2	115	13.2	754	86.8	
	Often	1406	40.7	211	15.0	1195	85.0	
	Rarely / Never	1178	34.1	196	16.6	982	83.4	
Social Activity								<0.0001
	Frequently	1071	31.0	180	16.8	891	83.2	
	Often	1985	57.5	231	11.6	1754	88.4	
	Rarely/Never	397	11.5	111	28.0	286	72.0	
Smoking								<0.0001
	Current smoker	727	21.1	71	9.8	656	90.2	
	Ex-smoker	575	16.7	65	11.3	510	88.7	
	Non-smoker	2151	62.3	386	17.9	1765	82.1	
Drinking								<0.0001
	Current drinker	1447	41.9	148	10.2	1299	89.8	
	Ex-drinker	397	11.5	69	17.4	328	82.6	
	Non-drinker	1609	46.6	305	19.0	1304	81.0	
Number of chronic diseases								<0.0001
	0	1503	43.5	160	10.6	1343	89.4	
	1	1096	31.7	176	16.1	920	83.9	
	2	569	16.5	123	21.6	446	78.4	
	3 or more	285	8.3	63	22.1	222	77.9	
Cancer								0.3611
	Yes	116	3.4	21	18.1	95	81.9	
	No	3337	96.6	501	15.0	2836	85.0	
Depression								<0.0001
	Yes	425	12.3	154	36.2	271	63.8	
	No	3028	87.7	368	12.2	2660	87.8	
Total		3453	100.0	522	15.1	2931	84.9	

† Low: MMSE ≤ 23, normal: 24 ≤ MMSE; ‡ Mini-Mental State Examination (MMSE); * The results of χ^2^ tests to analyze frequencies of each categorical variable by stress.

**Table 2 ijerph-17-01106-t002:** Generalized Linear Model (GEE) * with low cognitive function measured by MMSE ‡ score.

Variables	Low Cognitive Function †	
	OR	95% CI
General Quality of Life (QOL)		
Remained poor	1.33	(1.18–1.51)
Worsened	1.32	(1.18–1.47)
Improved	1.16	(1.03–1.30)
Remained good	1.00	
Financial aspect of QOL		
Remained poor	1.40	(1.23–1.58)
Worsened	1.37	(1.21–1.55)
Improved	1.25	(1.11–1.41)
Remained good	1.00	
Family aspect of QOL		
Remained poor	1.39	(1.23–1.58)
Worsened	1.33	(1.18–1.50)
Improved	1.19	(1.04–1.35)
Remained good	1.00	

† Low: MMSE ≤ 23, normal: 24 ≤ MMSE; ‡ Mini-Mental State Examination (MMSE); * Fully adjusted model.

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
