# Peer review of "The Association Between the Changes in General, Family, and Financial Aspects of Quality of Life and Their Effects on Cognitive Function in an Elderly Population: The Korean Longitudinal Study of Aging, 2008–2016"

_ijerph, 2020, doi:10.3390/ijerph17031106_

Round 1

Reviewer 1 Report

The paper covers a relevant topic, the relation between subjective, self-perceived aspect of ageing and cognitive functioning. Sadly, the methodological approach is questionable. Briefly:

The measurement of quality of life through single, analogical scales must be carefully explained and justified.  The hypothesis of the paper establish a causal relationship between changes in quality of life and changes in cognitive function. Sadly, the cross-sectional data analyzed do not allow to confirm causal relationships. I strongly suggest to include longitudinal analysis of the data, beyond (cognitively questionable) lag-time options. 

A better description of the sample according to their cognitive status, and differences between subsamples of the general study included or excluded in this paper, is also needed. 

Finally, a better description of the effect of covariates is recommended. 

Author Response

Dear Reviewer,

Thank you so very much for your comments as well as the outstanding suggestions and comments made by the reviewers. We have diligently reviewed each comment and have addressed the concerns and suggestions made. Below you will find detailed responses to each comment. We are excited for the potential to have this manuscript published by the International Journal of Environment and Public Health and welcome any further suggestions and/or comments.

Sincerely,

Wonjeong Chae (on behalf of Dr. Jang)

Reviewer Comment:

The paper covers a relevant topic, the relation between subjective, self-perceived aspect of ageing and cognitive functioning. Sadly, the methodological approach is questionable.

Briefly:

The measurement of quality of life through single, analogical scales must be carefully explained and justified. The hypothesis of the paper establish a causal relationship between changes in quality of life and changes in cognitive function. Sadly, the cross-sectional data analyzed do not allow to confirm causal relationships. I strongly suggest to include longitudinal analysis of the data, beyond (cognitively questionable) lag-time options.

Response:

Thank you for reviewing our manuscript. We appreciate your valuable comments. First, our variable, quality of life was one of the health indicators in our survey data. We chose to use that variable to self-reported from score 0 to 100 like EQ-VAS measure. And also, due to the survey format, there was no other question related to the quality of life so we used that single analogical scale measurement. Not like EQ-VAS, our quality of life score is actually given as the exact score, not like EQ-VAS, we believe that we obtained accuracy in perspective quality of life.

Second, related to the hypothesis of our study, we did not mean to discover the causal relationship between changes in quality of life and cognitive function. We realized that some of our wording caused confusion that we revised to deliver a clear message (Please see line 196-201 for the revised paragraph) We are sorry for phrasing wrongly. As you mentioned, within our data and study, we are not able to discover the causal relationship. We aimed to see an association between various aspects of quality of life and cognitive level. Because previous studies were mainly focused on health-related quality of care and cognitive function/level. Also, as this is a critical matter that we added in our limitation to state that we cannot show the causal relationship from the study. Please see line 275-278.

And, since our data is a longitudinal survey data, we were able to see that changes in quality of life. Therefore, we added lag-time to see the changes in quality of life for participants. As per the analysis, we used repeated measure analysis, the generalized estimating equation (GEE) for the longitudinal data even though our outcome is in the odds ratio. We revised the statistical analysis section with further details. Please see line 133-135.

A better description of the sample according to their cognitive status, and differences between subsamples of the general study included or excluded in this paper, is also needed.

Response:

Thank you for your valuable comment. For the study population, the first survey was conducted in 2006 with approximately 10,000 adults who are age 45 or over. We excluded participants who had dementia and a low cognitive level based on the MMSE score (less than 23). Therefore, in 2008, participants’ cognitive level was normal. We explained this in the below section for MMSE that we re-arranged this description into the study population. Then, we deleted participants who never been married in order to measure the familial aspect of the quality of health. Please revised the manuscript line 100-101.

Finally, a better description of the effect of covariates is recommended.

Response:

We appreciate your comments on our manuscript. Regarding the effect of covariates adjustment, we revised our manuscript line 125-126 as those covariates have been adjusted to control baseline variables that can impact the outcome especially the study with a large sample. Also, by adjusting covariates, we were able to improve precision of our study. However, our study might not include all factors that can have the effect the test result that we added in our limitations. Please see line 280.

Reviewer 2 Report

This manuscript has a very interesting topic regarding the potential relation between quality of life and cognitive impairment in elderly population. The mesure of quality of life is relatively easy to obtain, so it could be a useful tool for clinical professionals regarding cognitive impairment in old groups. However, in my opinion, there are some points that should be considered and arranged before the publication of the manuscript.

Introduction: the introduction section is very short. More references are needed to understand the importance of the topic. Specially, the references regarding the relationship between quality of life and cognitive impairment are not explained, only cited in two or three lines. Please, review the lines 64-81 and explain it with more details. It is important to understand the results of previous studies and its relation with the present manuscript.

Method: this section is, in general, well explained. However, I am worried about the cognitive level assessment. Authors used only one test, Mini-Mental State Examination. This is a good test for screening, but a poor measure of the real level of cognition of the participants. In my opinion, this is a very important limitation of the study, probably the most important one, and it is not mentioned anywhere. This weakness must be explained, justified and authors should explain how they can solve that in the rest of the paper.

Results: The results are, in general, well explained. However, there is a problem with Figure 1 and Figure 2. Figure 1 does not exist in the manuscript, there is only the title. Figure 2 is difficult to understand. More explanations of the figures are needed to understand the meaning and interpretation of them.

Discussion: The discussion is well detailed and explains better the relation with previous literature than the introduction. However, in the first paragraph (lines 175-180) the authors seem to explain the relation between the principal variables of the study in terms of causality. This paragraph must be rewrited, because between quality of life and cognitive level could be relation and covariation, but it is dangerous try to explain one and other in the actual terms. Maybe this was not the intention of the authors, and it is only a question of language. In any case, it is better to rewrite it.

Author Response

Reviewer Comment:

Introduction: the introduction section is very short. More references are needed to understand the importance of the topic. Specially, the references regarding the relationship between quality of life and cognitive impairment are not explained, only cited in two or three lines. Please, review the lines 64-81 and explain it with more details. It is important to understand the results of previous studies and its relation with the present manuscript.

Response:

Thank you for reviewing our manuscript. We appreciate your comments on our introduction. After reviewing your comment, our introduction needs more references that can explain the relationship between quality of life and cognitive impairment. As you mentioned, lines from 64 to 81, we revised and adder more information on those relationship. We described that quality of life could be one of measurement of health and among elder population and cognitive function is related to socioeconomic factors that are also related to quality of life. In addition, we mentioned that previous study focused on health-related quality of life yet, we focused on other aspects of quality of life such as familiar and financial aspects. Please see lines from 68-78 and line 83 in revised manuscript.

Method: this section is, in general, well explained. However, I am worried about the cognitive level assessment. Authors used only one test, Mini-Mental State Examination. This is a good test for screening, but a poor measure of the real level of cognition of the participants. In my opinion, this is a very important limitation of the study, probably the most important one, and it is not mentioned anywhere. This weakness must be explained, justified and authors should explain how they can solve that in the rest of the paper.

Response:

Thank you for your valuable comment. We appreciate your time and concerns regarding our manuscript. There were two reasons when we chose the Mini-Mental State Examination (MMSE) score to measure the cognitive level. First, our data is based on the survey and the MMSE score was one of the variable that was not self-reported data. We aimed to avoid self-reported data to investigated the cognitive level. Second, MMSE is a common examination that has been used in clinical settings to detect patients with cognitive problems, such as dementia. However, MMSE is not the most accurate and efficient indicator to measure the cognitive level. Therefore, as you commented, it has its weakness that we added in our limitation. Please see line 275-278.

Results: The results are, in general, well explained. However, there is a problem with Figure 1 and Figure 2. Figure 1 does not exist in the manuscript, there is only the title. Figure 2 is difficult to understand. More explanations of the figures are needed to understand the meaning and interpretation of them.

Response:

Thank you for your comment. We apologize for missing out on figure 2. There was an error during the formatting process. Those figures are the outcome from the subgroup analysis that we analyzed by gender and employment status. We added all figures correctly and provided further explanations related to the figures. Please see line 165-167 and figure 1 and figure 2.

Discussion: The discussion is well detailed and explains better the relation with previous literature than the introduction. However, in the first paragraph (lines 175-180) the authors seem to explain the relation between the principal variables of the study in terms of causality. This paragraph must be rewrited, because between quality of life and cognitive level could be relation and covariation, but it is dangerous try to explain one and other in the actual terms. Maybe this was not the intention of the authors, and it is only a question of language. In any case, it is better to rewrite it.

Response:

Thank you for reviewing our manuscript. After reviewing your comment, we agree that our writing could cause misunderstanding that we revised the first paragraph. We reworded the section to be more accurate. Please see line 196-201 in the revised manuscript and in limitation, line 279.

Reviewer 3 Report

The research results, including the range of values, are distractedly placed in the abstract. This makes the abstract confusing. It would be sufficient to state here what the results confirm or refute or just leave there conclusion only.

I would welcome at least an outline of a possible application.

Author Response

Reviewer Comment:

The research results, including the range of values, are distractedly placed in the abstract. This makes the abstract confusing. It would be sufficient to state here what the results confirm or refute or just leave there conclusion only.

I would welcome at least an outline of a possible application.

Response:

Thank you for reviewing our manuscript and we appreciate your comments. After receiving your comments and we do agree that our results in the Abstract could be described more sufficiently. Therefore, we revised the section by deleting 95% CI and mentioning those values were statistically significant. Please see line 26-29. Thank you.

Round 2

Reviewer 1 Report

Thank you for your review, which have improved your paper.

I would like to read a more carefully description of how the Generalized Linear Models are implemented. 

A proper definition of the covariates are still lacking (i.e. what do you mean with unemployment regarding employment status?).

A more detailed discussion about the source of gender differences (proper gender effect vs. cohort differences) would be valuable.  

Author Response

Thank you for your review, which have improved your paper.

Response: Thank you for reviewing our manuscript. Respect to your comments and suggestions, we revised our manuscript once again. Please see the revised manuscript. We deeply appreciate your consideration and support.

I would like to read a more carefully description of how the Generalized Linear Models are implemented. 

Response: Thank you for reviewing our manuscript. We revised with more description of our statistic analysis in line 138-141 in blue.

Revised as:
To analyze the longitudinal and repeated measures data using logistic regression would violate the assumption of independence. Therefore, we conducted a multivariable generalized estimating equation (GEE) model with a logit link and controlled for confounders.

A proper definition of the covariates are still lacking (i.e. what do you mean with unemployment regarding employment status?).

Response: Thank you for reviewing our manuscript. For your question, employment status is divided into two categories ‘employed’ and ‘unemployed’. I am afraid that you misunderstood our ‘unemployed’ category as unemployment status. Those categories are grouped as the survey question and answers. However, respect to your comment and provide easier understanding for readers, we revised our manuscript with each category's definition as needed. Please see line 128-135 in blue.

Revised as:
Factors were adjusted to control baseline variables that can have a possible effect on our study outcome and to improve the precision of the outcome. Therefore, in our study, factors related to demographics (sex, age), socioeconomics (education level, income level, employment status, marriage status, social activity: frequently – almost every day, once a week, twice a week, thrice a week; often – once a month, twice a month, once a two month; rarely – once a year, twice year, thrice a year), family (number of children, degree of face-to-face contact with children, degree of email or phone contact with children: frequently – almost every day, once a week, twice a week, thrice a week; often – once a month, twice a month, once a two month; rarely – once a year, twice year, thrice a year), health behavior (smoking, drinking) and health condition that diagnosed by their doctor (number of chronic diseases, cancer, depression) were adjusted for the study. All categories are grouped based on the survey question for those without further definition.

A more detailed discussion about the source of gender differences (proper gender effect vs. cohort differences) would be valuable.

Response: Thank you for your comment. In our discussion, we revised with additional references, we described gender effect and cohort difference. Please see line 230-243.

Revised as:
Our analyses support the theory of gender as a different mechanism in mental health [36,37] which can be presented as the gender effect. The study population includes individuals who are age 50 to 70 and above. Therefore, it might have cohort differences that shared different social and non-social events throughout the aging process. Despite the cohort effect on the elderly population, gender differences are easily found in a society that can possibly affect the quality of life and cognitive level [38,39,40,41]. Men and women tend to participate in different social activities which could influence their quality of life and/or cognitive level [39,40]. Regard to cognitive level, gender has a stronger association. Many studies have shown that women are at a higher risk of dementia and Alzheimer’s disease [14,40,41] and their study subjects were the elderly population of various ages. Their results were consistent with the investigations conducted on gender difference on dementia and Alzheimer’s disease among elders [14,40,41,42] and our results exhibited a higher OR of the low cognitive function groups in women. In addition, our study evaluated various aspects of quality of life and found that the effects of these aspects are different by gender [24,31,38] which is also interpreted as one part of gender effect.

Reviewer 2 Report

The modifications of the manuscript made it better for publication. However, I will suggest some aspects that I consider that could be improved and are necessary for its final publication:

Figures 1 and 2: there is a little explanation added regarding these figures. In my opinion, the meaning of the figures are still not clear and should be improved to understand them better.

Conclusion: the first line of the conclusion is, again, talking about causality. The discussion was modified but not the conclusion, and it must be in the same line, not talking in terms of causality.

Author Response

The modifications of the manuscript made it better for publication. However, I will suggest some aspects that I consider that could be improved and are necessary for its final publication:

Response: Thank you for reviewing our manuscript. Respect to your comments and suggestions, we revised our manuscript once again. Please see the revised manuscript. We deeply appreciate your consideration and support.

Figures 1 and 2: there is a little explanation added regarding these figures. In my opinion, the meaning of the figures are still not clear and should be improved to understand them better.

Response: Thank you for your valuable comment. We provided additional explanations related to figure 1 and 2. Please see 172-178 and 196-197 in blue. We changed the title of figures 1 and 2 to present clear outcomes and notify the value of the y-axis and footnotes on revised figures.

Revised as:
Subgroup analysis presents associations between quality of life and low cognitive level within the gender and the employment status of study the population. Figure 1 and Figure 2 presents the results of the subgroup analysis. Three aspects of quality of life are grouped individually and contain the degree of quality of life: remained poor, worsen, and improve, respectively.

Figure 1 contains the results of the subgroup analysis conducted by gender that shows low cognitive function measured by the MMSE score. Men and women express different aspects of quality of life regarding their cognitive function.

Figure 2 shows the results of subgroup analysis was conducted by employment status. The results demonstrated that participants who were unemployed had higher ORs of low cognitive function measured by the MMSE score.

Conclusion: the first line of the conclusion is, again, talking about causality. The discussion was modified but not the conclusion, and it must be in the same line, not talking in terms of causality.

Response: We appreciate your comments. After reviewing your comments, we revised the conclusion that is in line with the discussion. We replace the word ‘causal relationship’ to the association between quality of life and cognitive function. Please see page 307 in blue.

Revised as
We investigated the association between quality of life and cognitive function.